# A SIMPLE CONNECTION FROM LOSS FLATNESS TO COMPRESSED REPRESENTATIONS IN NEURAL NETWORKS

## ABSTRACT

Deep neural networks' generalization capacity has been studied in a variety of ways, including at least two distinct categories of approach: one based on the shape of the loss landscape in parameter space, and the other based on the structure of the representation manifold in feature space (that is, in the space of unit activities). These two approaches are related, but they are rarely studied together and explicitly connected. Here, we present a simple analysis that makes such a connection. We show that, in the last phase of learning of deep neural networks, compression of the manifold of neural representations correlates with the flatness of the loss around the minima explored by SGD. We show that this is predicted by a relatively simple mathematical relationship: flatter loss gives a lower upper bound on metrics of the compression of neural representations. Our results build on the prior work of Ma and Ying, which shows how flatness (i.e., small eigenvalues of the loss Hessian) develops in late phases of learning and leads to robustness to perturbations in network inputs. Moreover, we show there is no similarly direct connection between local dimensionality and sharpness, suggesting that this property may be controlled by different mechanisms than volume and hence may play a complementary role in neural representations. Overall, we advance a dual perspective on generalization in neural networks in both parameter and feature space.

## 1 INTRODUCTION

Deep neural networks' generalization capacity has been studied in many ways. Generalization is a complex phenomenon influenced by myriad factors, including model architecture, dataset size and diversity, and the specific task used to train a network. Researchers continue to develop new techniques to enhance generalization (Elsayed et al., 2018; Galanti et al., 2023). From a theoretical point of view, we can identify two distinct categories of approach. These are works that study neural network generalization in the context of (a) properties of minima of the loss function that learning algorithms find in parameter space (Dinh et al., 2017; Andriushchenko et al., 2023), and (b) properties of the representations that optimized networks find in feature space – that is, in the space of their neural activations (Ben-Shaul & Dekel, 2022; Ben-Shaul et al., 2023; Rangamani et al., 2023; Papyan et al., 2020).

One of the most widely studied factors that influence generalization is the shape of the loss landscape in parameter space. Empirical studies and theoretical analyses have shown that training deep neural networks using stochastic gradient descent (SGD) with a small batch size and a large learning rate often converges to flat and wide minima (Ma & Ying, 2021; Blanc et al., 2020; Geiger et al., 2021; Li et al., 2022; Wu et al., 2018; Jastrzebski et al., 2018; Xie et al., 2021; Zhu et al., 2019). Flat minima refer to regions in the loss landscape where the loss function has a relatively large basin: put simply, the loss doesn't change much in different directions around the minimum. Many works conjecture that flat minima lead to a simpler model (shorter description length), and thus are less likely to overfit and more likely to generalize well (Jastrzebski et al., 2018; Yang et al., 2023; Wu et al., 2018). However, whether flatness positively correlates with the network's generalization capability remains unsettled (Dinh et al., 2017; Andriushchenko et al., 2023; Yang et al., 2021). In particular, Dinh et al. (2017) argues that one can construct very sharp networks that generalize

well through reparametrization. However, more recent work (Andriushchenko et al., 2023) shows that even reparametrization-invariant sharpness cannot capture the relationship between sharpness and generalization.

In our work, we investigate how the sharpness of the loss function near learned solutions in parameter space influences local geometric features of neural representations. We demonstrate that as this sharpness decreases and the minima become flatter, there is a set of mathematical bounds that imply that the neural representation must undergo at least a specific, computable level of compression. This process, which is related to previous results including the concept of neural collapse (Farrell et al., 2022; Kothapalli et al., 2022; Zhu et al., 2021; Ansuini et al., 2019; Recanatesi et al., 2019; Papyan et al., 2020), refers to the emergence of a more compact and by some measures lower dimensional structure in the neural representation space. Compression in the feature space enables networks to isolate the most crucial and discriminative features of input data. As a model becomes less sensitive to small perturbations or noise in the input data, it gains increased robustness against variations between training and test data. This simple and direct relationship between compression and robustness creates a valuable lens into networks' potential to generalize.

We find that bounds that apply to two different metrics of compression – volumetric ratio and maximum local sensitivity – include different terms, and therefore predict different levels of compression for each. Moreover, we study the factors that contribute to the tightness of the bounds, or lack thereof – and hence may allow representations to display trends that in practice appear to contradict the theoretical predictions of the bounds. We also note that local dimensionality is a compression metric of a distinct nature, and therefore does not necessarily correlate with sharpness. Taken together, this reveals that the impact of loss function sharpness on the neural representation is more complex than a simple (and single) compression effect. These effects, despite their nuance, shed light on the complex link between sharpness and generalization.

Throughout, we focus on the second, or final, stage of learning, which proceeds after SGD has already found parameters that give near-optimal performance (i.e., zero training *error*) on the training data (Ma & Ying, 2021; Tishby & Zaslavsky, 2015; Ratzon et al., 2023). Here, additional learning still occurs, which changes the properties of the solutions in both feature and parameter space in very interesting ways.

Our work makes the following novel contributions:

- The paper identifies two representation space quantities that are bounded by sharpness – volume compression and maximum local sensitivity (MLS) – and gives new explicit formulas for these bounds that are reparametrization-invariant.

- The paper conducts empirical experiments with both VGG10 and MLP networks and finds that volume compression and MLS are indeed strongly correlated with sharpness.

- The paper finds that sharpness, volume compression, and MLS are also correlated, if more weakly, with test loss and hence generalization.

In these ways, we help reveal the interplay between key properties of trained neural networks in parameter space and representation space. Specifically, we identify a sequence of equality conditions for the bounds that link the volume and MLS of the neural representations to the sharpness in parameter space. These conditions are helpful in explaining why there are the mixed results on the relationship between sharpness and generalization in the literature, by looking through the additional lens of the induced representations. Our findings altogether suggest that allied views into representation space offer a valuable dual perspective to that of parameter space landscapes for understanding the effects of learning on generalization.

Our paper proceeds as follows. First, we review arguments of Ma & Ying (2021) that flatter minima can constrain the gradient of the loss with respect to network inputs and extend the formulation to the multidimensional input case (Sec. 2). Next, we prove that lower sharpness implies a lower upper bound on two metrics of the compression of the representation manifold in feature space: the local volume and the maximum local sensitivity (MLS) (Sec. 3.1, Sec. 3.2). We conclude our findings with simulations that confirm our central theoretical results and show how they can be applied in practice (Sec. 4).

## 2 BACKGROUND AND SETUP

Consider a feedforward neural network $f$ with input data $\mathbf{x} \in \mathbb{R}^M$ and parameters $\boldsymbol{\theta}$. The output of the network is:

$$\mathbf{y} = f(\mathbf{x}; \boldsymbol{\theta}) , \tag{1}$$

with and $\mathbf{y} \in \mathbb{R}^N$ ($N < M$). Consider a quadratic loss $L(\mathbf{y}, \mathbf{y}_{\text{true}}) = \frac{1}{2}||\mathbf{y} - \mathbf{y}_{\text{true}}||^2$ function of the outputs and ground truth $\mathbf{y}_{\text{true}}$. In the following, we'll simply write $L(\mathbf{y})$, $L(f(\mathbf{x}, \boldsymbol{\theta}))$ or simply $L(\boldsymbol{\theta})$ to highlight the dependence of the loss on the output, the network or its parameters.

During the last phase of learning, Ma and colleagues have recently argued that SGD appears to regularize the sharpness of the loss (Li et al., 2022) (see also (Wu et al., 2018; Jastrzebski et al., 2018; Xie et al., 2021; Zhu et al., 2019)). This is to say that the dynamics of SGD lead network parameters to minima where the local loss landscape is flatter or wider. This is best captured by the sharpness, measured by the sum of the eigenvalues of the Hessian:

$$S(\boldsymbol{\theta}) = \text{Tr}(H) , \tag{2}$$

with $H = \nabla^2 L(\boldsymbol{\theta})$ being the Hessian. A solution with low sharpness is a flatter solution. Following (Ma & Ying, 2021; Ratzon et al., 2023), we define $\boldsymbol{\theta}^*$ to be an "exact interpolation solution" on the zero training loss manifold in the parameter space (the zero loss manifold in what follows), where $f(\mathbf{x}_i, \boldsymbol{\theta}^*) = \mathbf{y}_i$ for all $i$'s (with $i \in \{1..n\}$ indexing the training set) and $L(\boldsymbol{\theta}^*) = 0$. On the zero loss manifold, in particular, we have

$$S(\boldsymbol{\theta}^*) = \frac{1}{n} \sum_{i=1}^n \|\nabla_{\boldsymbol{\theta}} f(\mathbf{x}_i, \boldsymbol{\theta}^*)\|_F^2 \tag{3}$$

where $\|\cdot\|_F$ is the Frobenius norm. We give the proof of this equality in Appendix A. In practice, the parameter $\boldsymbol{\theta}$ will never reach an exact interpolation solution due to the gradient noise of SGD, however, Eq. (3) is a good enough approximation of the sharpness as long as we find an approximate interpolation solution (Lemma. A.1).

In order to see why minimizing the sharpness of the solution leads to more compressed representations, we need to move from parameter space to input space. To do so we review the argument of Ma & Ying (2021) that relates variations in input data $\mathbf{x}$ and input weights. Let $\mathbf{W}$ be the input weights (the parameters of the first linear layer) of the network, and $\bar{\boldsymbol{\theta}}$ the rest of the parameters. Following (Ma & Ying, 2021), as the weights $\mathbf{W}$ multiply the inputs $\mathbf{x}$ we have the following identities:

$$\|\nabla_{\mathbf{W}} f(\mathbf{W}\mathbf{x}; \bar{\boldsymbol{\theta}})\|_F = \sqrt{\sum_{i,j,k} J_{jk}^2 x_i^2} = \|J\|_F \|\mathbf{x}\|_2 \geq \|J\|_2 \|\mathbf{x}\|_2$$
$$\nabla_{\mathbf{x}} f(\mathbf{W}\mathbf{x}; \bar{\boldsymbol{\theta}}) = \mathbf{W}^T J , \tag{4}$$

where $J = \frac{\partial f(\mathbf{W}\mathbf{x}; \bar{\boldsymbol{\theta}})}{\partial(\mathbf{W}\mathbf{x})}$ is a complex expression as computed in, e.g., backpropagation. From Eq. (4) and the sub-multiplicative property of the Frobenius norm and the matrix 2-norm, we have:

$$\|\nabla_{\mathbf{x}} f(\mathbf{W}\mathbf{x}; \bar{\boldsymbol{\theta}})\|_F \leq \frac{\|\mathbf{W}\|_F}{\|\mathbf{x}\|_2} \|\nabla_{\mathbf{W}} f(\mathbf{W}\mathbf{x}; \bar{\boldsymbol{\theta}})\|_F ,$$
$$\|\nabla_{\mathbf{x}} f(\mathbf{W}\mathbf{x}; \bar{\boldsymbol{\theta}})\|_2 \leq \frac{\|\mathbf{W}\|_2}{\|\mathbf{x}\|_2} \|\nabla_{\mathbf{W}} f(\mathbf{W}\mathbf{x}; \bar{\boldsymbol{\theta}})\|_F. \tag{5}$$

If the norms $\|\mathbf{W}\|_F$ or $\|\mathbf{W}\|_2$ and $\|\mathbf{x}\|_2$ are not excessively large or small respectively, these bounds control the gradient with respect to inputs via the gradient with respect to weights. This in turn reveals the impact of flatness in the loss function:

$$\frac{1}{n} \sum_{i=1}^n \|\nabla_{\mathbf{x}} f(\mathbf{x}_i, \boldsymbol{\theta}^*)\|_F^k \leq \frac{\|\mathbf{W}\|_F^k}{\min_i \|\mathbf{x}_i\|_2^k} \frac{1}{n} \sum_{i=1}^n \|\nabla_{\mathbf{W}} f(\mathbf{x}_i, \boldsymbol{\theta}^*)\|_F^k$$
$$\leq \frac{\|\mathbf{W}\|_F^k}{\min_i \|\mathbf{x}_i\|_2^k} \frac{1}{n} \sum_{i=1}^n \|\nabla_{\boldsymbol{\theta}} f(\mathbf{x}_i, \boldsymbol{\theta}^*)\|_F^k. \tag{6}$$

125 We define $G := \frac{1}{n}\sum_{i=1}^{n}\|\nabla_{\mathbf{x}}f(\mathbf{x}_i,\boldsymbol{\theta}^*)\|_F^2$ when $k = 2$. Similarly,

$$\frac{1}{n}\sum_{i=1}^{n}\|\nabla_{\mathbf{x}}f(\mathbf{x}_i,\boldsymbol{\theta}^*)\|_2^k \le \frac{\|\mathbf{W}\|_2^k}{\min_i\|\mathbf{x}_i\|_2^k}\frac{1}{n}\sum_{i=1}^{n}\|\nabla_{\boldsymbol{\theta}}f(\mathbf{x}_i,\boldsymbol{\theta}^*)\|_F^k. \tag{7}$$

126 Thus, in (Ma & Ying, 2021), the effect of input perturbations is constrained by the sharpness of the
127 loss function. The flatter the minimum of the loss, the lower the effect of input space perturbations
128 on the network function $f(\mathbf{x},\boldsymbol{\theta}^*)$ as determined by gradients.

129 ## 3  FROM ROBUSTNESS TO INPUTS TO COMPRESSION OF REPRESENTATIONS

130 We now further analyze variations in the input and how they propagate through the network to shape
131 representations of sets of inputs. Although we only study the representations of the output of the
132 network here, our results apply to representations of any middle layer, through defining $f$ to be
133 the transformation from input to the middle layer of interest. Ovearll, we focus on 3 key metrics
134 of network representations: local dimensionality, volumetric ratio, and maximum local sensitivity.
135 These quantities enable us to establish and evaluate the influence of input variations and, in turn,
136 sharpness on neural representation properties.

137 ### 3.1  WHY SHARPNESS BOUNDS LOCAL VOLUMETRIC TRANSFORMATION IN
138         REPRESENTATION SPACE

139 Consider an input data point $\bar{\mathbf{x}}$ drawn from the training set: $\bar{\mathbf{x}} = \mathbf{x}_i$ for a specific $i \in \{1..n\}$. Let
140 the set of all possible perturbations around $\bar{\mathbf{x}}$ in input space be the ball $\mathcal{B}(\bar{\mathbf{x}})_\alpha \sim \mathcal{N}(\bar{\mathbf{x}}, \alpha\mathcal{I})$, where
141 $\alpha$ depends on the perturbation's covariance, given as $C_{\mathcal{B}(\mathbf{x})} = \alpha\mathcal{I}$, with $\mathcal{I}$ as the identity matrix.
142 We'll explore the network's representation of inputs by measuring the expansion or contraction of
143 the ball $\mathcal{B}(\bar{\mathbf{x}})_\alpha$ as it propagates through the network. We first propagate the ball through the network
144 transforming each point $\mathbf{x}$ into its image $f(\mathbf{x})$. Following a Taylor expansion for points within
145 $\mathcal{B}(\bar{\mathbf{x}})_\alpha$ as $\alpha \to 0$ we have:

$$f(\mathbf{x}) = f(\bar{\mathbf{x}}) + \nabla_{\mathbf{x}}(f(\bar{\mathbf{x}},\boldsymbol{\theta}^*))(\mathbf{x} - \bar{\mathbf{x}}). \tag{8}$$

146 We can express the limit of the covariance matrix $C_{f(\mathcal{B}(\mathbf{x}))}$ of the output $f(\mathbf{x})$ as

$$C_f^{\lim} := \lim_{\alpha\to 0} C_{f(\mathcal{B}(\mathbf{x})_\alpha)} = \alpha\nabla_{\mathbf{x}}f(\bar{\mathbf{x}},\boldsymbol{\theta}^*)\nabla_{\mathbf{x}}^T f(\bar{\mathbf{x}},\boldsymbol{\theta}^*), \tag{9}$$

147 Our covariance expressions capture the distribution of points in $\mathcal{B}(\bar{\mathbf{x}})_\alpha$ as they go through the net-
148 work $f(\bar{\mathbf{x}},\boldsymbol{\theta}^*)$.

149 Now we quantify how a network compresses its input volumes via the local volumetric ratio, be-
150 tween an hypercube of side length $h$ at $\mathbf{x}$ and its image under transformation $f$:

$$d\,\mathrm{Vol}^{ratio}|_{f(\mathbf{x},\boldsymbol{\theta}^*)} = \lim_{h\to 0}\frac{\mathrm{Vol}(f(\mathbf{x},\boldsymbol{\theta}^*))}{\mathrm{Vol}(\mathbf{x})}$$
$$= \sqrt{\det\left(\nabla_{\mathbf{x}}f^T\nabla_{\mathbf{x}}f\right)} \tag{10}$$

151 which is equal to the square root of the product of all positive eigenvalues of $C_f^{\lim}$. Exploiting the
152 bound on the gradients derived earlier in Eq. (5), we derive a similar bound for the volumetric ratio:

$$d\,\mathrm{Vol}^{ratio}|_{f(\mathbf{x},\boldsymbol{\theta}^*)} \le \left(\frac{\mathrm{Tr}\,\nabla_{\mathbf{x}}f^T\nabla_{\mathbf{x}}f}{N}\right)^{N/2}$$
$$= N^{-N/2}\|\nabla_{\mathbf{x}}f(\mathbf{x},\boldsymbol{\theta}^*)\|_F^N \tag{11}$$

153 where the first line uses the inequality between arithmetic and geometric means and the second the
154 definition of the Frobenius norm. Introducing the averaged volumetric ratio across all input points
155 $dV^{ratio}(\boldsymbol{\theta}^*) = \frac{1}{n}\sum_{i=1}^{n} d\,\mathrm{Vol}^{ratio}|_{f(\mathbf{x}_i,\boldsymbol{\theta}^*)}$, we obtain:

$$dV^{ratio}(\boldsymbol{\theta}^*) \le \frac{N^{-N/2}}{n}\sum_{i=1}^{n}\|\nabla_{\mathbf{x}}f(\mathbf{x},\boldsymbol{\theta}^*)\|_F^N \le \frac{n^{\max(N/2-1,0)}\|\mathbf{W}\|_F^N}{\min_i\|\mathbf{x}_i\|_2^N}\left(\frac{S(\boldsymbol{\theta}^*)}{N}\right)^{N/2}. \tag{12}$$

for all $N \geq 1$. A detailed derivation of the above inequality is given in Appendix B. Eq. (12) implies that flattened minima of the loss function in parameter space contribute to the compression of the data's representation manifold. Our analysis demonstrates that these two phenomena are linked by the robustness properties of the network to input perturbations.

## 3.2 MAXIMUM LOCAL SENSITIVITY AS AN ALLIED METRIC TO TRACK NEURAL REPRESENTATION GEOMETRY

We observe that the equality condition in the first line of Eq. (11) rarely holds in practice, since to achieve equality, we need all singular values of the Jacobian matrix $\nabla_{\mathbf{x}} f$ to be identical. Our experiments in Sec. 4 show that the local dimensionality decreases rapidly with training onset, indicating that $\nabla_{\mathbf{x}} f^T \nabla_{\mathbf{x}} f$ has a non-uniform eigenspectrum. Moreover, the volume will decrease rapidly as the smallest eigenvalue vanishes. Thus, although sharpness upper bounds the volumetric ratio, it does not correlate well with the it, nor does volumetric ratio give an accurate estimate of sharpness. Fortunately, considering only the maximum eigenvalue instead of the product alleviates this problem (recall that $\det\left(\nabla_{\mathbf{x}} f^T \nabla_{\mathbf{x}} f\right)$ in the definition Eq. (10) or volumetric ratio is the product of all eigenvalues): we define the maximum local sensitivity (MLS) to be the largest singular value of $\nabla_{\mathbf{x}} f$. The MLS is equivalently the matrix 2-norm of $\nabla_{\mathbf{x}} f$. Intuitively, it is the largest possible local change of $f(\mathbf{x})$ when the norm of the perturbation to $\mathbf{x}$ is regularized. We denote the sample mean of MLS as $\overline{\text{MLS}}$. Given this definition, we obtain a bound of MLS using the Frobenius norm of the first linear layer, the quadratic mean of the input norm, and the sharpness.

$$\overline{\text{MLS}} = \frac{1}{n} \sum_{i=1}^{n} \|\nabla_{\mathbf{x}} f(\mathbf{x}_i, \boldsymbol{\theta}^*)\|_2 \leq \|\mathbf{W}\|_2 \sqrt{\frac{1}{n} \sum_{i=1}^{n} \frac{1}{\|\mathbf{x}_i\|_2^2}} S(\boldsymbol{\theta}^*)^{1/2} . \tag{13}$$

The derivation of the above bound is included in Appendix C, where we use Cauchy-Swartz inequality to tighten the bound in Eq. (7). As an alternative measure of compressed representations, we empirically show in Appendix D.2 that MLS has higher correlation with sharpness and test loss than the other two measures we consider in the feature space. We include more analysis of the tightness of this bound in Appendix D and discuss its connection to other works therein.

## 3.3 LOCAL DIMENSIONALITY IS TIED TO, BUT NOT BOUNDED BY, SHARPNESS

Now we introduce a local measure of dimensionality based on this covariance, the local Participation Ratio, given by:

$$D_{\text{PR}}(f(\bar{\mathbf{x}})) = \lim_{\alpha \to 0} \frac{\text{Tr}[C_{f(\mathcal{B}(\mathbf{x}))}]^2}{\text{Tr}[(C_{f(\mathcal{B}(\mathbf{x}))})^2]} = \frac{\text{Tr}[C_f^{\text{lim}}]^2}{\text{Tr}[(C_f^{\text{lim}})^2]} \tag{14}$$

(cf. (Gao et al., 2017; Litwin-Kumar et al., 2017; Recanatesi et al., 2022)). This quantity can be averaged across a set of samples: $D_{\text{PR}}(\theta^*) = \frac{1}{n} \sum_{i=1}^{n} D_{\text{PR}}(f(\mathbf{x}_i))$. This quantity in some sense represents the sparseness of the eigenvalues of $C_f^{\text{lim}}$: if we let $\boldsymbol{\lambda}$ be all the eigenvalues of $C_f^{\text{lim}}$, then the local dimensionality can be written as $D_{\text{PR}} = (\|\boldsymbol{\lambda}\|_1 / \|\boldsymbol{\lambda}\|_2)^2$, which attains its maximum value when all eigenvalues are equal to each other, and its minimum when all but one eigenvalue is non-zero. Note that the quantity retains the same value when $\boldsymbol{\lambda}$ is arbitrarily scaled, therefore it is hard to find a relationship between local dimensionality and $\|\nabla_{\mathbf{x}} f(\mathbf{x}, \boldsymbol{\theta}^*)\|_F^2$, which is basically $\|\boldsymbol{\lambda}\|_1$.

# 4 EXPERIMENTS

## 4.1 SHARPNESS AND COMPRESSION: VERIFYING THE THEORY

The theoretical results derived above show that during the later phase of training – the interpolation phase – measures of compression of the network's representation is upper bounded by a function of the sharpness of the loss function in parameter space. This links sharpness and compression of representation: the flatter is the loss landscape, the lower is the upper bound on the representation's compression metrics.

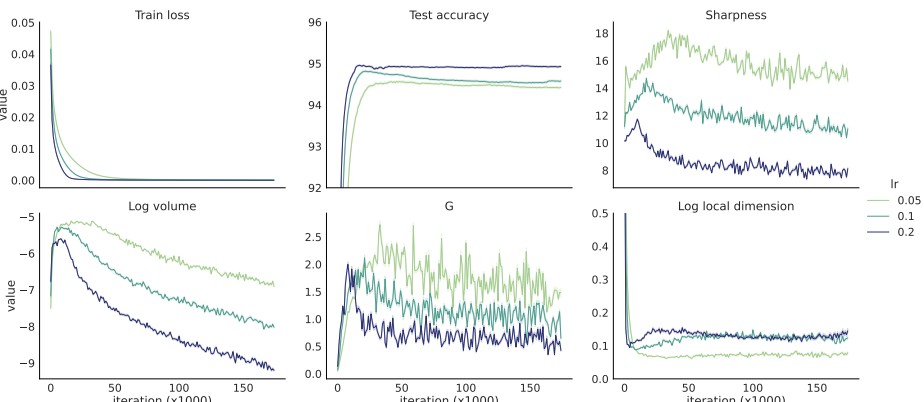

Figure 1: Trends in key variables across SGD training of the VGG10 network with fixed batch size (equal to 20) and varying learning rates (0.05, 0.1 and 0.2). After the loss is minimized (so that an approximate interpolation solution is found) sharpness and volumes decrease together. Moreover, higher learning rates lead to lower sharpness and hence stronger compression. From left to right: train loss, test accuracy, sharpness (square root of Eq. (3)), log volumetric ratio (Eq. (10)), left-hand side of Eq. (6) with $k = 2$ (axes titled $G$), and local dimensionality of the network output (Eq. (14)).

It remains to test in practice, however, whether these bounds are sufficiently tight so that a clear relationship between sharpness and representation collapse appears. As one such test, we ran the following experiment. We trained a network (Simonyan & Zisserman, 2015) to classify images from the CIFAR-10 dataset, and calculated the sharpness (Eq. (2)), the log volumetric ratio (Eq. (10)) and the left-hand side of Eq. (6) (the gradient with respect to the inputs, a quantity we term G in the figures below) during the training phase (Fig 1 and 2). We trained the network (VGG10) using SGD on images from 2 classes (out of 10) so that convergence to the interpolation regime, i.e. zero error, was faster. We explored the influence of two specific parameters that have a substantial effect on the network's training: learning rate and batch size. For each pair of learning rate and batch size parameters, we computed all quantities at hand across 100 input samples and five different random initializations for network weights.

In the first set of experiments, we studied the link between a decrease in sharpness during the latter phases of training and volume compression (Fig. 1). We noticed that when the network reaches the interpolation regime, and the sharpness decreases, so does the volume. The quantity G similarly decreases. All these results were consistent across multiple learning rates for a fixed batch size (of 20): specifically, for learning rates that gave lower values of sharpness, volume was lower as well.

We then repeated the experiments while keeping the learning rate fixed (lr=0.1) and varying the batch size. The same broadly consistent trends emerged linking a decrease in the sharpness to a compression in the representation volume (Fig. 2). However, we also find that while sharpness stops decreasing after about iteration $50 \cdot 10^3$ for batch size 32, the volume keeps decreasing as learning proceeds. This suggests that there may be other mechanisms at play, beyond sharpness, in driving the compression of volumes.

We repeat the experiments with an MLP trained on the FashionMNIST dataset (Fig. E.8 and Fig. E.7). Although the sharpness does not noticeably decrease at the end of the training, the sharpness has the same trend as G, which is consistent with our bound. The volume keeps decreasing after the sharpness plateaus, but it is also decreasing at a much slower rate, again matching our theory while suggesting that an additional factor is also involved in its decrease.

## 4.2 SHARPNESS AND COMPRESSION ON TEST SET DATA

Even though Eq. (3) is exact for interpolation solutions only (i.e., those with zero loss), we found that the test loss is small enough (Fig. 3) so that it should be a good approximation for test data as well. Therefore we analyzed our simulations to study trends in sharpness and volume for these held-out test data as well (Fig. 3). We discovered that this sharpness increased rather than diminished as

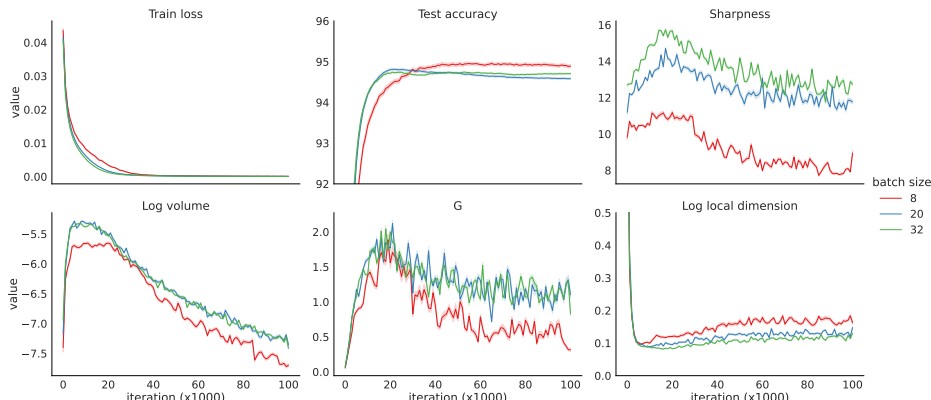

Figure 2: Trends in key variables across SGD training of the VGG10 network with fixed learning rate size (equal to 0.1) and varying batch size (8, 20, and 32). After the loss is minimized (so that an interpolation solution is found) sharpness and volumes decrease together. Moreover, lower batch sizes lead to lower sharpness and hence stronger compression. From left to right in row-wise order: train loss, test accuracy, sharpness (square root of Eq. (3)), log volumetric ratio (Eq. (10)), left-hand side of Eq. (6) with $k = 2$ (axes titled $G$), and local dimensionality of the network output (Eq. (14)).

a result of training. We hypothesized that sharpness could correlate with the difficulty of classifying testing points. This was supported by the fact that the sharpness of misclassified test data was even greater than that of all test data. Again we see that G has the same trend as the sharpness. Despite this increase in sharpness, the volume followed the same pattern as the training set. This suggests that compression in representation space is a robust phenomenon that can be driven by additional phenomena beyond sharpness. Nevertheless, the compression still is weaker for misclassified test samples that have higher sharpness than other test samples. Overall, these results emphasize an interesting distinction between how sharpness evolves for training vs. test data.

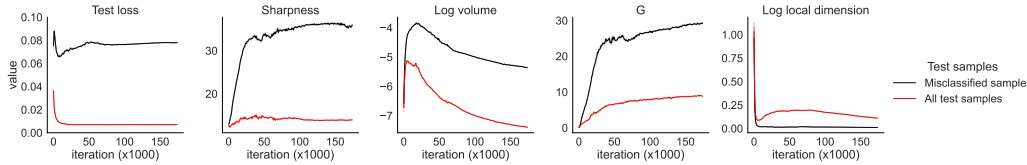

Figure 3: Trends in key variables across SGD training of the VGG10 network with fixed learning rate (equal to 0.1) and batch size (equal to 20) for samples of the test set. After the loss is minimized, we compute sharpness and volume on the test set. Moreover, the same quantities are computed separately over the entire test set or only on samples that are misclassified. In order from left to right in row-wise order: train loss, test loss, sharpness (Eq. (2)), log volumetric ratio (Eq. (10)), left-hand side of Eq. (6) with $k = 2$ (axes titled $G$), and local dimensionality of the network output (Eq. (14)).

### 4.3 SHARPNESS AND LOCAL DIMENSIONALITY

Lastly, we analyze the representation's local dimensionality in a manner analogous to the analysis of volume and MLS. A priori, it is ambiguous whether the dimensionality of the data representation should increase or decrease as the volume is compressed. For instance, the volume could decrease while maintaining its overall form and symmetry, thus preserving its dimensionality. Alternatively, one or more of the directions in the relevant tangent space could be selectively compressed, leading to an overall reduction in dimensionality.

Figures 1 and 2 show our experiments computing the local dimensionality over the course of learning. Here, we find that the local dimensionality of the representation decreases as the loss decreases to near 0, which is consistent with the viewpoint that the network compresses representations in

feature space as much as possible, retaining only the directions that code for task-relevant features (Berner et al., 2020; Cohen et al., 2020). However, the local dimensionality exhibits unpredictable behavior that cannot be explained by the sharpness once the network is near the zero-loss manifold and training continues. This discrepancy is consistent with the bounds established by our theory, which only bound the numerator of Eq. (14). It is also consistent with the property of local dimensionality that we described in Sec. 3.3 overall: it encodes the sparseness of the eigenvalues but it does not encode the magnitude of them. This shows how local dimensionality is a distinct quality of network representations compared with volume, and is driven by mechanisms that differ from sharpness alone. We emphasize that the dimensionality we study here is a local measure, on the finest scale around a point on the "global" manifold of unit activities; dimension on larger scales (i.e., across categories or large sets of task inputs (Farrell et al., 2022; Gao et al., 2017)) may show different trends.

## 5 CONCLUSION

This work presents a dual perspective, uniting views in both parameter and in feature space, of several key properties of trained neural networks that have been linked to their ability to generalize. We identify two representation space quantities that are bounded by sharpness – volume compression and maximum local sensitivity – and give new explicit formulas for these bounds. We conduct experiments with both VGG10 and MLP networks and find that the predictions of these bounds are born out for these networks, illustrating how MLS in particular is strongly correlated with sharpness. We also establish that sharpness, volume compression, and MLS are correlated, if more weakly, with test loss and hence generalization. Overall, we establish explicit links between sharpness properties in parameter spaces and compression and robustness properties in representation space.

By demonstrating both how these links can be tight, and how and when they may also become loose, we show that taking this dual perspective can bring more clarity to the often confusing question of what quantifies how well a network will generalize in practice. Indeed, many works, as reviewed in the introduction, have demonstrated how sharpness in parameter space can lead to generalization, but recent studies have established contradictory results. We show how looking at quantities not only in the parameter space (sharpness), but also in the feature space (compression, maximum local sensitivity, etc.) may help explain the wide range of results.

This said, we view our study as a starting point to open doors between two often-distinct perspectives on generalization in neural networks. Additional theoretical and experimental research is warranted to systematically investigate the implications of our findings, with a key area being further learning problems, such as predictive learning, beyond the classification tasks studied here. Nevertheless, we are confident that highly interesting and clarifying findings lie ahead at the interface between the parameter and representation space quantities explored here.

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

## A   PROOF OF EQ. (3)

**Lemma A.1.** *If $\boldsymbol{\theta}$ is an approximate interpolation solution, i.e. $\|f(\mathbf{x}_i, \boldsymbol{\theta}) - \mathbf{y}_i\| < \varepsilon$ for $i \in \{1, 2, \cdots, n\}$, and second derivatives of the network function $\|\nabla_{\theta_j^2} f(\mathbf{x}_i, \boldsymbol{\theta})\| < M$ is bounded, then*

$$S(\boldsymbol{\theta}^*) = \frac{1}{n} \sum_{i=1}^{n} \|\nabla_{\boldsymbol{\theta}} f(\mathbf{x}_i, \boldsymbol{\theta}^*)\|_F^2 + O(\varepsilon) \tag{15}$$

*Proof.* Using basic calculus we get

$$S(\boldsymbol{\theta}) = \text{Tr}(\nabla^2 L(\boldsymbol{\theta}))$$

$$= \frac{1}{2n} \sum_{i=1}^{n} \text{Tr}(\nabla_{\boldsymbol{\theta}}^2 \|f(\mathbf{x}_i, \boldsymbol{\theta}) - \mathbf{y}_i\|^2)$$

$$= \frac{1}{2n} \sum_{i=1}^{n} \text{Tr} \, \nabla_{\boldsymbol{\theta}} (2(f(\mathbf{x}_i, \boldsymbol{\theta}) - \mathbf{y}_i)^T \nabla_{\boldsymbol{\theta}} f(\mathbf{x}_i, \boldsymbol{\theta}))$$

$$= \frac{1}{n} \sum_{i=1}^{n} \sum_{j=1}^{m} \frac{\partial}{\partial \boldsymbol{\theta}_j} ((f(\mathbf{x}_i, \boldsymbol{\theta}) - \mathbf{y}_i)^T \nabla_{\boldsymbol{\theta}} f(\mathbf{x}_i, \boldsymbol{\theta}))_j$$

$$= \frac{1}{n} \sum_{i=1}^{n} \sum_{j=1}^{m} \frac{\partial}{\partial \boldsymbol{\theta}_j} (f(\mathbf{x}_i, \boldsymbol{\theta}) - \mathbf{y}_i)^T \nabla_{\boldsymbol{\theta}_j} f(\mathbf{x}_i, \boldsymbol{\theta})$$

$$= \frac{1}{n} \sum_{i=1}^{n} \sum_{j=1}^{m} \|\nabla_{\boldsymbol{\theta}_j} f(\mathbf{x}_i, \boldsymbol{\theta})\|_2^2 + (f(\mathbf{x}_i, \boldsymbol{\theta}) - \mathbf{y}_i)^T \nabla_{\boldsymbol{\theta}_j}^2 f(\mathbf{x}_i, \boldsymbol{\theta})$$

$$= \frac{1}{n} \sum_{i=1}^{n} \|\nabla_{\boldsymbol{\theta}} f(\mathbf{x}_i, \boldsymbol{\theta})\|_F^2 + \frac{1}{n} \sum_{i=1}^{n} (f(\mathbf{x}_i, \boldsymbol{\theta}) - \mathbf{y}_i)^T \nabla_{\boldsymbol{\theta}_j}^2 f(\mathbf{x}_i, \boldsymbol{\theta}).$$

Therefore

$$\left| S(\boldsymbol{\theta}) - \frac{1}{n} \sum_{i=1}^{n} \|\nabla_{\boldsymbol{\theta}} f(\mathbf{x}_i, \boldsymbol{\theta})\|_F^2 \right| < \frac{1}{n} \sum_{i=1}^{n} |(f(\mathbf{x}_i, \boldsymbol{\theta}) - \mathbf{y}_i)^T \nabla_{\boldsymbol{\theta}_j}^2 f(\mathbf{x}_i, \boldsymbol{\theta})| < M\varepsilon = O(\varepsilon). \tag{16}$$

$\square$

In other words, when the network reaches zero training error and enters the interpolation phase (i.e. it classifies all training data correctly), Eq. (3) will be a good enough approximation of the sharpness because the quadratic training loss is sufficiently small.

## B   PROOF OF EQ. (12)

We first show that Eq. (6) is correct. Because of Eq. (5), we have the first inequality of Eq. (6),

$$\frac{1}{n} \sum_{i=1}^{n} \|\nabla_{\mathbf{x}} f(\mathbf{x}_i, \boldsymbol{\theta}^*)\|_F^k \leq \|\mathbf{W}\|_F^k \frac{1}{n} \sum_{i=1}^{n} \frac{\|\nabla_{\mathbf{w}} f(\mathbf{x}_i, \boldsymbol{\theta}^*)\|_F^k}{\|\mathbf{x}_i\|_2^k}$$

$$\leq \frac{\|\mathbf{W}\|_F^k}{\min_i \|\mathbf{x}_i\|_2^k} \frac{1}{n} \sum_{i=1}^{n} \|\nabla_{\mathbf{w}} f(\mathbf{x}_i, \boldsymbol{\theta}^*)\|_F^k. \tag{17}$$

Since the input weights $\mathbf{W}$ is just a part of all the weights ($\boldsymbol{\theta}$) of the network, we have $\|\nabla_{\mathbf{w}} f(\mathbf{x}_i, \boldsymbol{\theta}^*)\|_F^k \leq \|\nabla_{\boldsymbol{\theta}} f(\mathbf{x}_i, \boldsymbol{\theta}^*)\|_F^k$. Therefore

$$\frac{\|\mathbf{W}\|_F^k}{\min_i \|\mathbf{x}_i\|_2^k} \frac{1}{n} \sum_{i=1}^{n} \|\nabla_{\mathbf{w}} f(\mathbf{x}_i, \boldsymbol{\theta}^*)\|_F^k \leq \frac{\|\mathbf{W}\|_F^k}{\min_i \|\mathbf{x}_i\|_2^k} \frac{1}{n} \sum_{i=1}^{n} \|\nabla_{\boldsymbol{\theta}} f(\mathbf{x}_i, \boldsymbol{\theta}^*)\|_F^k. \tag{18}$$

To show the correctness of Eq. (12), we discuss two cases.

**Case 1:** $k \geq 2$

**Lemma B.1.** *For vector* $\mathbf{x}$, $\|\mathbf{x}\|_p \geq \|\mathbf{x}\|_q$ *for* $1 \leq p \leq q \leq \infty$.

*Proof.* First we show that for $0 < k < 1$, we have $(|a| + |b|)^k \leq |a|^k + |b|^k$. It's trivial when either $a$ or $b$ is 0. So W.L.O.G, we can assume that $|a| < |b|$, and divide both sides by $|b|^k$. Therefore it suffices to show that for $0 < t < 1$, $(1+t)^k < t^k + 1$. Let $f(t) = (1+t)^k - t^k - 1$, then $f(0) = 0$, and $f'(t) = k(1+t)^{k-1} - kt^{k-1}$. Because $k - 1 < 0$, $1+t > 1$ and $t < 1$, $t^{k-1} > (1+t)^{k-1}$. Therefore $f'(t) < 0$ and $f(t) < 0$ for $0 < t < 1$. Combining all cases, we have $(|a| + |b|)^k \leq |a|^k + |b|^k$ for $0 < k < 1$. By induction, we have $(\sum_n |a_n|)^k \leq \sum_n |a_n|^k$.

Now we can prove the lemma using the conclusion above,

$$\left(\sum_n |x_n|^q\right)^{1/q} = \left(\sum_n |x_n|^q\right)^{p/q \cdot 1/p} \leq \left(\sum_n (|x_n|^q)^{p/q}\right)^{1/p} = \left(\sum_n |x_n|^p\right)^{1/p} \quad (19)$$

$\square$

Now take the $x_i$ in above lemma to be $\|\nabla_{\boldsymbol{\theta}} f(\mathbf{x}_i, \boldsymbol{\theta}^*)\|_F^2$ and let $p = 1, q = k/2$, then we get

$$\left(\sum_{i=1}^n (\|\nabla_{\boldsymbol{\theta}} f(\mathbf{x}_i, \boldsymbol{\theta}^*)\|_F^2)^{k/2}\right)^{2/k} \leq \sum_{i=1}^n \|\nabla_{\boldsymbol{\theta}} f(\mathbf{x}_i, \boldsymbol{\theta}^*)\|_F^2. \quad (20)$$

Therefore,

$$\frac{\|\mathbf{W}\|_F^k}{\min_i \|\mathbf{x}_i\|_2^k} \frac{1}{n} \sum_{i=1}^n \|\nabla_{\boldsymbol{\theta}} f(\mathbf{x}_i, \boldsymbol{\theta}^*)\|_F^k \leq \frac{n^{k/2-1} \|\mathbf{W}\|_F^k}{\min_i \|\mathbf{x}_i\|_2^k} \left(\frac{1}{n} \sum_{i=1}^n \|\nabla_{\boldsymbol{\theta}} f(\mathbf{x}_i, \boldsymbol{\theta}^*)\|_F^2\right)^{k/2}$$
$$= \frac{n^{k/2-1} \|\mathbf{W}\|_F^k}{\min_i \|\mathbf{x}_i\|_2^k} S(\boldsymbol{\theta}^*)^{k/2} \quad (21)$$

**Case 2:** $1 \leq k < 2$

**Lemma B.2.** *For vector* $\mathbf{x} \in \mathbb{R}^n$, $\|\mathbf{x}\|_p \leq n^{1/p-1/q} \|\mathbf{x}\|_q$ *for* $1 \leq p \leq q \leq \infty$.

*Proof.* By Hölder's inequality, we have,

$$\sum_i |x_i|^p = \sum_i |x_i|^p \cdot 1 \leq \left(\sum_i |x_i|^q\right)^{p/q} \left(\sum_i 1\right)^{1-p/q} = n^{1-p/q} \|\mathbf{x}\|_q^p \quad (22)$$

Taking the p-th root on both sides gives us the desired inequality. $\square$

Now take the $x_i$ in above lemma to be $\|\nabla_{\boldsymbol{\theta}} f(\mathbf{x}_i, \boldsymbol{\theta}^*)\|_F$ and let $p = k, q = 2$, then we get

$$\left(\sum_{i=1}^n (\|\nabla_{\boldsymbol{\theta}} f_i\|_F)^k\right)^{1/k} \leq n^{1/k-1/2} \left(\sum_{i=1}^n \|\nabla_{\boldsymbol{\theta}} f_i\|_F^2\right)^{1/2}. \quad (23)$$

Therefore,

$$\frac{\|\mathbf{W}\|_F^k}{\min_i \|\mathbf{x}_i\|_2^k} \frac{1}{n} \sum_{i=1}^n \|\nabla_{\boldsymbol{\theta}} f_i\|_F^k \leq \frac{\|\mathbf{W}\|_F^k}{\min_i \|\mathbf{x}_i\|_2^k} \frac{n^{1-k/2}}{n} \left(\sum_{i=1}^n \|\nabla_{\boldsymbol{\theta}} f_i\|_F^2\right)^{k/2}$$
$$= \frac{\|\mathbf{W}\|_F^k}{\min_i \|\mathbf{x}_i\|_2^k} S(\boldsymbol{\theta}^*)^{k/2}. \quad (24)$$

Combining Eq. (21) and Eq. (24), we get Eq. (12).

## C    PROOF OF EQ. (13)

From Eq. (5), we get

$$\overline{\mathrm{MLS}} = \frac{1}{n}\sum_{i=1}^{n}\|\nabla_{\mathbf{x}}f_i\|_2 \leq \|\mathbf{W}\|_2\frac{1}{n}\sum_{i=1}^{n}\frac{\|\nabla_{\mathbf{W}}f(\mathbf{x}_i,\boldsymbol{\theta}^*)\|_F}{\|\mathbf{x}_i\|_2}. \tag{25}$$

Now Cauchy Swartz inequality tells us that

$$\left(\sum_{i=1}^{n}\frac{\|\nabla_{\mathbf{W}}f_i\|}{\|\mathbf{x}_i\|_2}\right)^2 \leq \left(\sum_{i=1}^{n}\frac{1}{\|\mathbf{x}_i\|^2}\right)\cdot\left(\sum_{i=1}^{n}\|\nabla_{\mathbf{W}}f_i\|^2\right). \tag{26}$$

Therefore

$$\overline{\mathrm{MLS}} \leq \|\mathbf{W}\|_2\sqrt{\frac{1}{n}\sum_{i=1}^{n}\frac{1}{\|\mathbf{x}_i\|^2}}\cdot\sqrt{\frac{1}{n}\sum_{i=1}^{n}\|\nabla_{\mathbf{W}}f_i\|^2}$$

$$\leq \|\mathbf{W}\|_2\sqrt{\frac{1}{n}\sum_{i=1}^{n}\frac{1}{\|\mathbf{x}_i\|^2}}\cdot S(\boldsymbol{\theta}^*)^{1/2}. \tag{27}$$

## D    EMPIRICAL ANALYSIS OF THE BOUND

### D.1    TIGHTNESS OF THE BOUND

In this section, we mainly explore the tightness of the bound in Eq. (13) for reasons discussed in Sec. 3.2. First we rewrite Eq. (13) as

$$\overline{\mathrm{MLS}} = \frac{1}{n}\sum_{i=1}^{n}\|\nabla_{\mathbf{x}}f(\mathbf{x}_i,\boldsymbol{\theta}^*)\|_2 \qquad := A$$

$$\leq \frac{\|\mathbf{W}\|_2}{n}\sum_{i=1}^{n}\frac{\|\nabla_{\mathbf{W}}f(\mathbf{x}_i,\boldsymbol{\theta}^*)\|_F}{\|\mathbf{x}_i\|_2} \qquad := B$$

$$\leq \|\mathbf{W}\|_2\sqrt{\frac{1}{n}\sum_{i=1}^{n}\frac{1}{\|\mathbf{x}_i\|_2^2}}\sqrt{\frac{1}{n}\sum_{i=1}^{n}\|\nabla_{\mathbf{W}}f(\mathbf{x}_i,\boldsymbol{\theta}^*)\|_F^2} \qquad := C \tag{28}$$

$$\leq \|\mathbf{W}\|_2\sqrt{\frac{1}{n}\sum_{i=1}^{n}\frac{1}{\|\mathbf{x}_i\|_2^2}}S(\boldsymbol{\theta}^*)^{1/2} \qquad := D$$

Thus Eq. (13) consists of 3 different steps of relaxations. We analyze them one by one:

1. $(A \leq B)$ The equality holds when $\|W^T J\|_2 = \|W\|_2\|J\|_2$ and $\|J\|_F = \|J\|_2$, where $J = \frac{\partial f(\mathbf{Wx};\bar{\boldsymbol{\theta}})}{\partial(\mathbf{Wx})}$. The former equality requires that $W$ and $J$ have the same left singular vectors. The latter requires $J$ to have zero singular values except for the largest singular value. Since $J$ depends on the specific neural network architecture and training process, we test the tightness of this bound empirically (Fig. D.4).

2. $(B \leq C)$ The equality requires $\frac{\|\nabla_{\mathbf{W}}f(\mathbf{x}_i,\boldsymbol{\theta}^*)\|_F}{\|\mathbf{x}_i\|_2}$ to be the same for all $i$. In other words, the bound is tight when $\frac{\|\nabla_{\mathbf{W}}f(\mathbf{x}_i,\boldsymbol{\theta}^*)\|_F}{\|\mathbf{x}_i\|_2}$ does not vary too much from sample to sample.

3. $(C \leq D)$ The equality holds if the model is linear, i.e. $\boldsymbol{\theta} = \mathbf{W}$.

We empirically verify the tightness of the above bounds in Fig. D.4

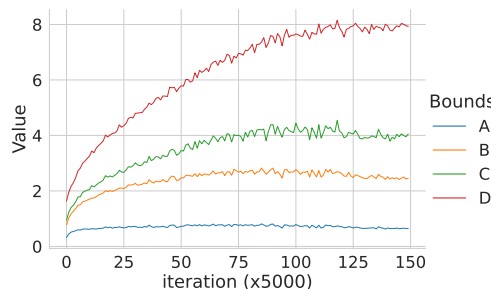

Figure D.4: **Empirical tightness of the bounds.** We empirically verify that the inequalities in Eq. (28) hold and test their tightness. The results are shown for a fully connected feedforward network trained on the FashionMNIST dataset. The quantities A, B, C, and D are defined in Eq. (28). We see that the gap between C and D is large compared to the gap between A and B or B and C. This indicates that partial sharpness $\|\nabla_{\mathbf{W}} f(\mathbf{x}_i, \boldsymbol{\theta}^*)\|_F$ (sensitivity of the loss w.r.t. only the input weights) is more indicative of the change in the maximum local sensitivity (A). Indeed, correlation analysis shows that bound C is positively correlated with MLS while bound D, perhaps surprisingly, is negatively correlated with MLS (Fig. D.6).

## D.2 CORRELATION ANALYSIS

We empirically show how different metrics correlate with each other, and how these correlations can be predicted from our bounds. We train 20 VGG10 networks with different batch sizes, learning rates, and random initialization to classify images from the CIFAR-10 dataset, and plot pairwise scatter plots between 5 quantities at the end of the training: test loss, MLS, G (see Eq. (6)), log volume, sharpness and local dimensionality (Fig. D.5).

We find that

1. G and MLS are highly correlated and can be almost seen as the same quantity, scaled.

2. Although the bound in Eq. (12) is loose, log volume correlates well with sharpness and MLS.

3. Sharpness is positively correlated with the test loss, indicating that little reparametrization effect (Dinh et al., 2017) is happening during training, i.e. the network weights do not change too much during training. This is consistent with observations in Ma & Ying (2021).

4. MLS improves the correlation with the test loss over log volume and local dimensionality. This is consistent with the bound Eq. (13).

We repeat the analysis on an MLP trained on the FashionMNIST dataset, and observe the same phenomena (Fig. D.6).

## D.3 CONNECTION TO OTHER WORKS

Our bound and its analysis are connected to many theoretical and experimental results. First of all, the right-hand side of Eq. (13) is related not only to the sharpness but also to the norm of the input weights. Therefore our bound takes into the effect of reparametrization, and is invariant under scaling of the input weights. This is consistent with the theoretical results in Dinh et al. (2017) which show that sharpness can be arbitrarily increased by reparametrization while the network can still generalize. Moreover, many works studied simplified linear models (Li et al., 2022; Ding et al., 2023; Nacson et al., 2022; Gatmiry et al., 2023), and showed that the flattest minima generalize well. Correspondingly, Eq. (28) shows that when the neural network is linear, the inequality between C and D becomes equality, and the flattest minima give the tightest bound on MLS. On the other hand, this also explains why sharpness does not always correlate with generalization when the network becomes more complicated (Wen et al., 2023; Andriushchenko et al., 2023): having weights that are other than the input weights makes the bound looser and more unpredictable. Experiments on

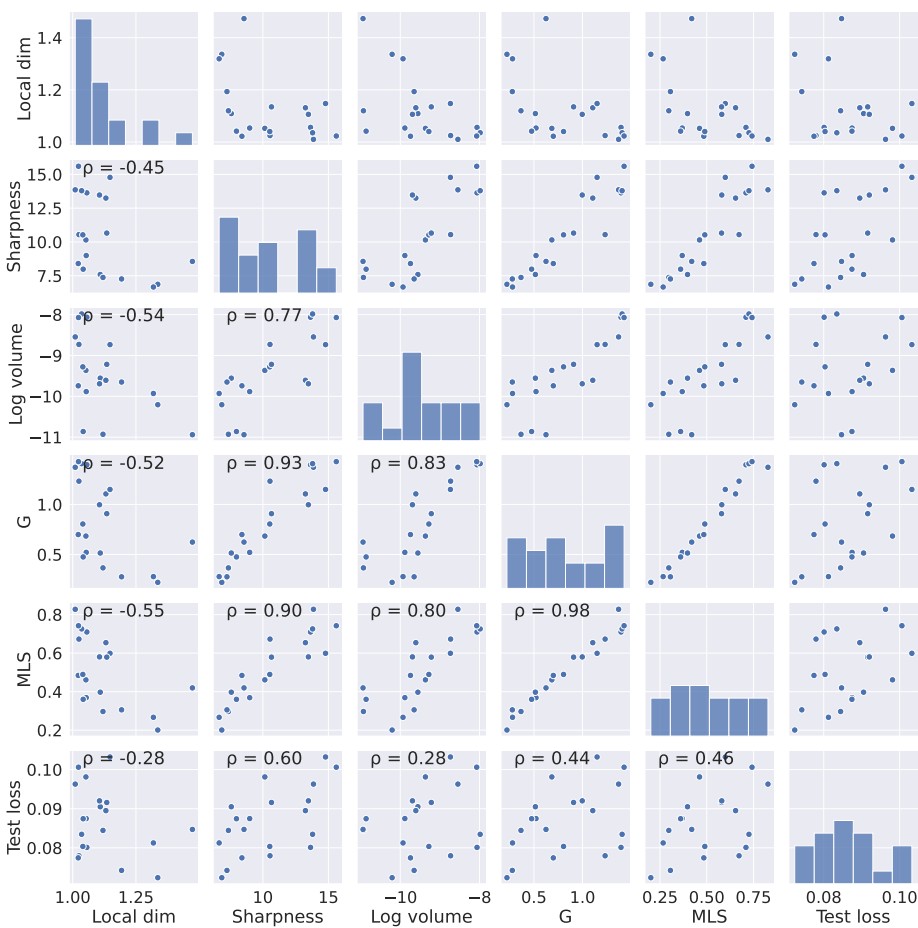

Figure D.5: **Pairwise correlation among different metrics.** We trained 20 different VGG10 networks using vanilla SGD with different learning rates, batch sizes, and random initializations and plot pairwise scatter plots between different quantities: local dimensionality, sharpness (square root of Eq. (3)), log volume (Eq. (10)), G (Eq. (6)), MLS (Eq. (13)) and test loss. The Pearson correlation coefficient $\rho$ is shown in the top-left corner for each pair of quantities. See Appendix D.2 for a summary of the findings in this figure.

MLP show that the bound D in Eq. (28) can even be negatively correlated with MLS and test loss (Fig. D.6).

# E ADDITIONAL EXPERIMENTS

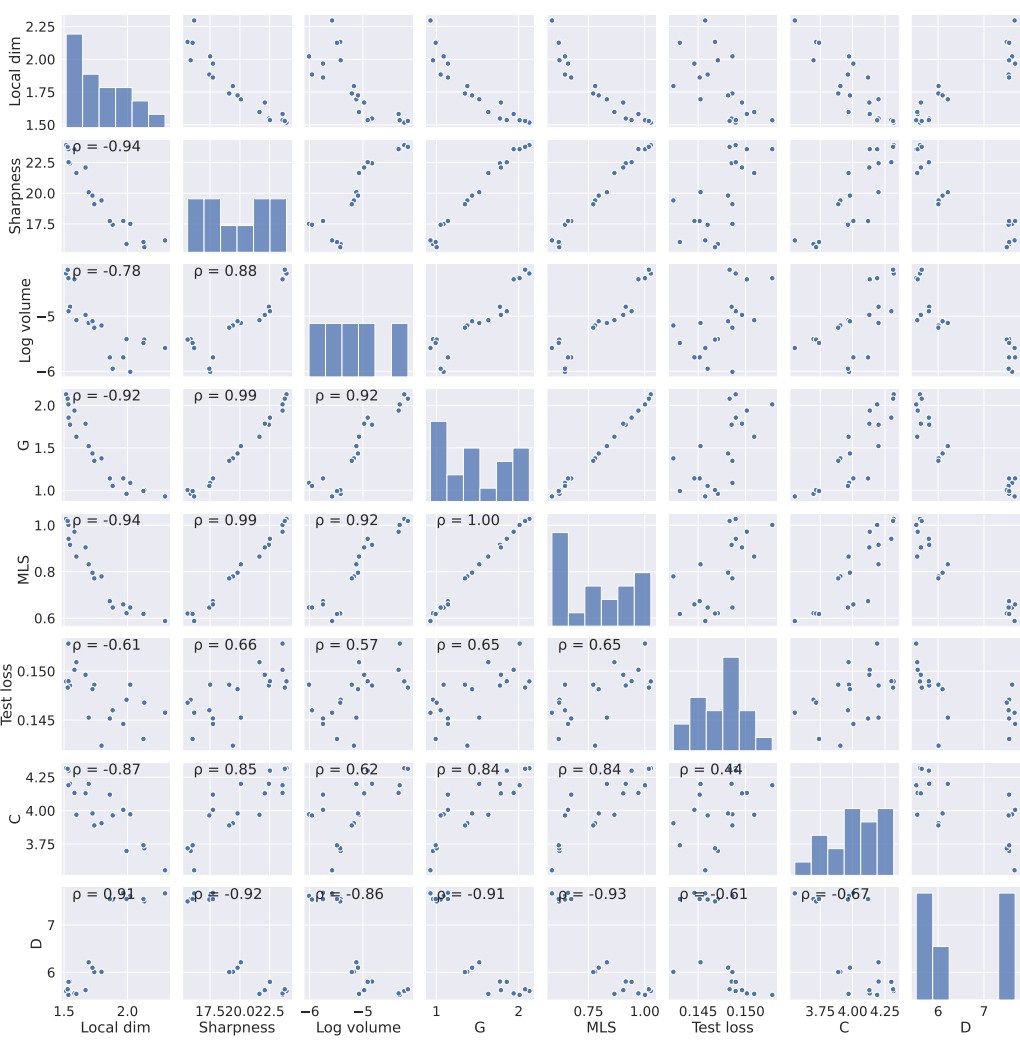

Figure D.6: **Pairwise correlation among different metrics.** We trained 20 different 4-layer MLPs using vanilla SGD with different learning rates, batch size, and random initializations and plot pairwise scatter plots between different quantities: local dimensionality, sharpness (square root of Eq. (3)), log volume (Eq. (10)), G (Eq. (6)), MLS (Eq. (13)), test loss and additionally bound C and D as defined in Eq. (28). The Pearson correlation coefficient $\rho$ is shown in the top-left corner for each pair of quantities. See Appendix D.2 for a summary of the findings in this figure.

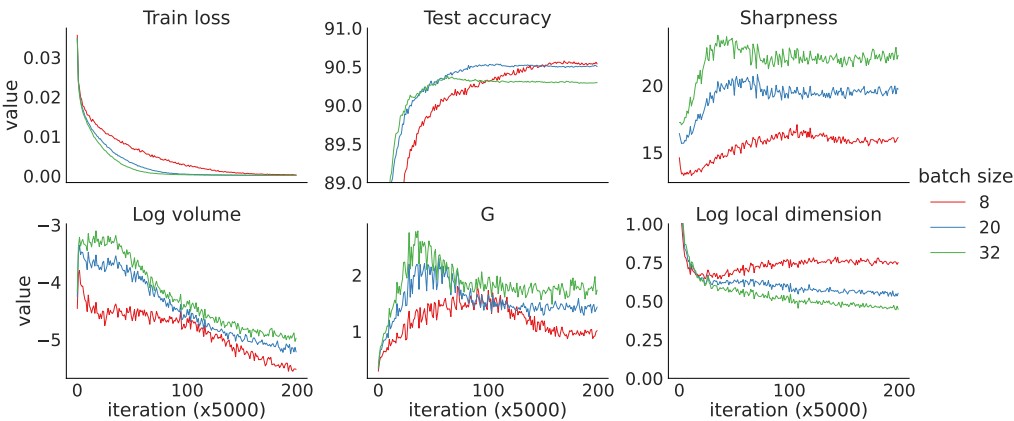

Figure E.7: Trends in key variables across SGD training of a 4-layer MLP with fixed learning rate (equal to 0.1) and varying batch size (8, 20, and 32). After minimizing the loss, lower batch sizes lead to lower sharpness and stronger compression. Moreover, G closely follows the trend of sharpness during the training. From left to right: train loss, test accuracy, sharpness (square root of Eq. (3)), log volumetric ratio (Eq. (10)), G (Eq. (6)), and local dimensionality of the network output (Eq. (14)).

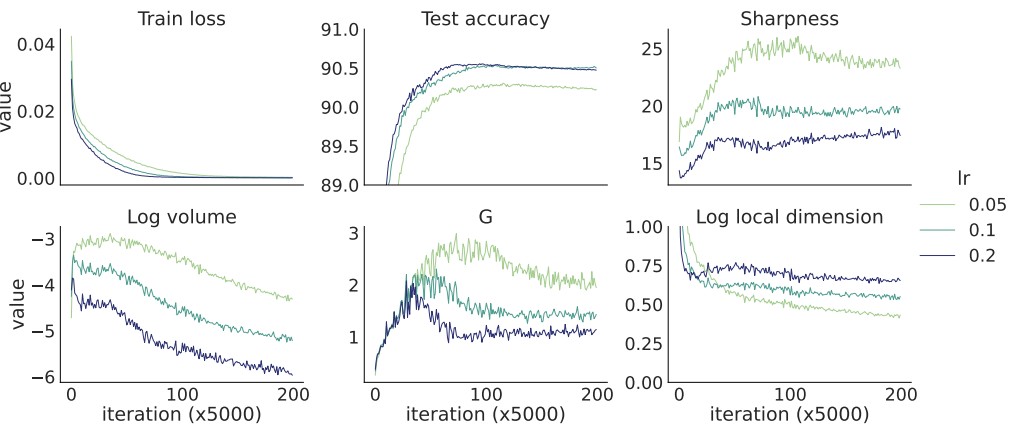

Figure E.8: Trends in key variables across SGD training of a 4-layer MLP with fixed batch size (equal to 20) and varying learning rates (0.05, 0.1 and 0.2). After the loss is minimized, higher learning rates lead to lower sharpness and hence stronger compression. Moreover, G closely follows the trend of sharpness during the training. From left to right: train loss, test accuracy, sharpness (square root of Eq. (3)), log volumetric ratio (Eq. (10)), G (Eq. (6)), and local dimensionality of the network output (Eq. (14)).

