# OpenReview forum: "A simple connection from loss flatness to compressed representations in neural networks"
_ICLR.cc/2024/Conference — Submitted to ICLR 2024_

### Official Review · Reviewer_s2iN · 2023-10-27

**Soundness:** 1 poor
**Presentation:** 1 poor
**Contribution:** 1 poor
**Rating:** 3
**Confidence:** 4

**Summary:**

The idea of the paper is to connect flatness as a generalization metric and representation compression (like in neural collapse phenomena). The authors point out that these two metrics are investigated as indicators of good generalization, but there were not connected to each other before. The authors concentrate on the interpolation regime when a neural network has reached minimal loss and derive a boundary on particular form of volume of the representations with a particular measure of flatness. An empirical investigation on VGG10 and CIFAR10 dataset demonstrated the reduction in volume and (mostly) sharpness.

**Strengths:**

Connection of flatness, as well as compression (in a sense of neural collapse conjecture) to the generalization abilities of models is an important research area. Interconnection between possible generalization metrics can prove to be useful in deriving unified understanding of generalization in neural networks.

**Weaknesses:**

The paper gives an impression of an unfinished work. It is not clearly signified which of the derivations are novel contributions and which are reproducing the existing research. The experimental evaluation is very limited and not fully showing the correctness/tightness of the bound presented.

Even with the assumption that trace of the Hessian is indeed a good generalization indicator (which was disproven by Dinh et al.), the discussion in the beginning of section 2 is very imprecise. If the loss indeed reaches the point of zero value and its derivatives are zero, SGD would not change the solution anymore - just by definition. The referenced works take into account label noise for example, that allows for further dynamic of the optimization even after reaching a minimal loss. Therefore the theoretical justification of the bounds is ill-posed and it also does not connect to the experiments performed, since there network there does not achieve 0 loss and 0 gradients (which should effectively stop the changes of the model).

The observations made in section 4.2 are basically showing that the derived boundary is extremely imprecise, which to some extent diminishes its value.

Minor:

- the positioning of images makes it very hard to follow the text and description of the experiments, they should be moved closer to the part where they are described

- short mention of possible perspective through the lens of entropy (in the end of section3) should be either elaborated more or removed, because currently it is much more confusing

- eq. 14 is introduced in the very end of the paper, while being used in all the experimental section which makes reading unnecessarily hard

**Questions:**

1 - What model of SGD dynamics do you have in mind when using terms "second phase" and "interpolation regime"?

2 - How the used volume connects to the representation collapse?

3 - What is the additional metric introduced in the last section of the paper?

---

> ### Author Response · Authors · 2023-11-22
> **Thank you**
>
> We appreciate the reviewer’s critiques and suggestions. Below we address the reviewer’s concerns point by point:
>
> Weakness:
>
> Novelty:  While we acknowledge and appreciate the reviewer’s point that our contributions could and should be better described, our impression from other reviewers is that the central novelty of newly connecting the previously separate ideas in parameter and representation space was laid out reasonably.  This said we embrace the criticism and the chance to do better.  First, in our revision, we have now clearly divided the material of the paper into two distinct sections. Sec. 2 is titled “Background and setup” and Sec. 3 highlights our original contributions in making new connections to neural representation space.  Second, also following the reviewer’s other very on-point suggestions, we have extended these original contributions in Sec. 3 significantly, with three new figures there and the allied appendix D (please see below).  Third, we now enumerate our contributions more explicitly in the introduction.  Overall, we believe that this has significantly improved our paper, and again thank the reviewer for their keen criticism and the chance to respond.
>
> Correctness/tightness of the bound: We thank the reviewer for their keen and on-target point that the tightness of the bound should be both more fully explored and presented.  In our revision, we have extended our analysis to tackle this important matter in two separate steps. First, we studied very carefully the tightness of our newly derived bound (Sec. 3.2 Eq. 13) in Appendix D.1, including isolating and testing three factors that contribute to this tightness or the lack thereof (with full details given in the new Appendix D.1, with a series of new equations and a new figure). Second, we now clearly connect the relationship between our bound and the insights derived from allied research (Appendix D.3). Third, we carry out and present new experiments on a different network (a fully connected MLP) (Fig D.6, E.7-8). Altogether our extended findings reveal both how our bound can predict that neural representations will be compressed, and also the factors that would limit the success of this prediction.  Further, dedicated future work will be required to assess when and how flatness increases or decreases through network training. We further detail our findings, again inspired by the reviewer's critique, with two new figures (Appendix D.2) that show how in our experiments a decrease in sharpness (and related increase in flatness) significantly correlated with volumetric compression of the neural representation.
>
> Imprecise argument in section 2: We thank the reviewer for pointing out the lack of precision of the argument at the start of section 2. We addressed this in two separate ways: 1) We now evaluate in more depth whether sharpness decreases towards the end of training. For example, sharpness does not necessarily decrease after the training error becomes zero (Fig. E.7-8). However, our additional experiments show that our bounds successfully predict the correlation between sharpness and both local volume and maximum local sensitivity (MLS) (Fig. D.5-6). 2) Our theoretical derivation and results do not rely on the specific form of label noise SGD or sharpness-aware minimization (SAM). Both the narrative and results of the paper are largely unaltered. The mathematical argument also has a general form that does not strictly rely on these specific forms of gradient noise ($\delta\theta^*$). We believe that it provides a constructive argument towards why flatness may/may not increase at the end of training (sharpness may not decrease when the noise is too small, i.e. the second term $\delta\theta^* H\delta\theta^*$ is negligible). Nevertheless, we omit this intuition from the paper itself to retain the focus on our main contributions. We believe these two points address the reviewer's concern.
>
> We believe that our updated Sec 3 and Appendix D now explain the looseness of the original bound that is evidenced by results in Appendix D. In summary, we found that the reason for the looseness in Eq. 12 is due to the sparseness of the eigenspectrum of $C_f^{lim}$. We again observe that in Fig. 3, although log volume is not well correlated with sharpness, the quantity G is much better correlated with sharpness. Exploiting this, we introduce a new, closely related metric, the maximum local sensitivity (MLS), that allows for a much tighter bound (Sec. 3.2). We then newly empirically test and show the correctness and tightness of this bound (Appendix D.1).
>
> Minor:
> 1. We have updated the figure positions to where the figure is mentioned.
> 2. We have removed the short mention of possible perspective through the lens of entropy as suggested.
> 3. We have moved (previous) Eq. 14 and the corresponding definition of local dimensionality to Sec 3 where all the metrics in the feature space are defined.

---

> ### Author Response · Authors · 2023-11-22
> **Response to questions**
>
> Questions:
> 1. All of our experiments use vanilla SGD. We would like to reiterate that our main contribution has no dependence on which optimization scheme is used.
>
> 2. Neural collapse is a phenomenon that occurs in deep neural networks when the training error reaches zero and the network enters the terminal phase of training. At this point, the within-class variability of final hidden layer outputs becomes infinitesimally small. This means that the distribution collapses down to a point or a very compressed manifold. The volume of the representation in neural space, which we study here, is one of the metrics that can capture the fact that such within-class variability is small.
>
> 3. The metric is called participation ratio which is a natural continuous measure of the local dimensionality.  Intuitively if all variance is concentrated in one direction, i.e. the covariance matrix $C_f^{lim}$ only has one non-zero eigenvalue, then the output of the network is locally 1D given input of a small ball B(x) around x. This is a metric that captures the neural compression effect through the lens of the dimensionality of the manifold of within-class representations. The participation ratio overall has been recently applied fairly broadly in the computational neuroscience and neural networks literature, making us confident that its analysis extends the value of the paper; moreover, we have updated the manuscript to include a dedicated subsection 3.1 to more completely describe and discuss this metric.

---

> > ### Comment · Reviewer_s2iN · 2023-12-05
> >
> > I appreciate the work that the authors performed to improve the paper. I believe my points were sufficiently addressed, but by changing more than 50% of the paper and introducing new focus of the contribution.
> >
> > I would highly encourage the authors to resubmit the paper and keep the improved formulations, but the current changes made for the rebuttal were significantly more than usual minor changes and the paper needs to be peer-reviewed anew.

---

### Official Review · Reviewer_Km7j · 2023-10-27

**Soundness:** 3 good
**Presentation:** 3 good
**Contribution:** 3 good
**Rating:** 8
**Confidence:** 3

**Summary:**

The authors study the interplay between 1) the flatness of the loss landscape at minima found during training, and 2) dimensionality of learned neural representations.  Building on Ying and Ma, the authors prove that flatter minima in the loss imply an upper bound on the volume occupied by neural representations. Experiments on a (simplified) image classification task confirm their analysis.

**Strengths:**

Originality:

The authors draw an original connection between two major concepts: flatness of loss landscapes and compression of representations.  These ideas have been studied independently quite thoroughly, but this thread is, to my knowledge, new.  Further, their experimental analysis bolsters support to this thread.

Quality/clarity:

The proofs and empirical studies were, to my eye, rigorous, clear, and thorough.
The paper is clearly structured, easy to follow, and provides intuitive explanations.
The authors clearly acknowledge the limitations of their work.

Significance:

This work is another solid contribution to the ongoing research thread of trying to understand the connection/interplay between flatness/sharpness and generalization.

**Weaknesses:**

Not that this work is particularly weak, but there could probably be more empirical analysis.  I'd love to see the analysis repeated for additional learning problems, beyond the simple classification problem studied.

**Questions:**

No questions.

---

> ### Author Response · Authors · 2023-11-22
> **Thank you**
>
> We appreciate the reviewer’s overall assessment of the value of our central contribution in linking flatness and compressed representations.  We also appreciate their stated limitation and suggestion, and next, describe how we have revised and added to our paper in response:
>
> Weakness:
>
> We agree with the reviewer’s suggestion that more empirical analysis should be included. In our revision, we have added new experiments on a different network (MLP) trained on the FashionMNIST dataset (Fig. D6, E7-8).  We have also added a new experimental Appendix D that comprehensively analyzes the tightness of the derived bound in a new figure, as well as another new figure showing correlations between different metrics introduced in the paper. We also agree that further extensions to other learning problems would be of interest, and added a sentence describing this in the revised discussion; we feel that completing these studies here would go beyond the scope of what we can accomplish within the given paper but are eager to undertake this in future studies.

---

### Official Review · Reviewer_Hwwc · 2023-10-29

**Soundness:** 3 good
**Presentation:** 3 good
**Contribution:** 3 good
**Rating:** 6
**Confidence:** 5

**Summary:**

The paper explores the correlation between the sharpness of minima and the corresponding representation manifold volume. The authors show, theoretically as well as experimentally, that in the last stage of training, when the loss is zero, further training leads to (1) flatter minima, which correspond to (2) a more compressed representation of inputs. As such, authors provide further theoretical and empirical support for an existing understanding of generalisation behaviour in neural networks.

**Strengths:**

This is a well-written paper on a very relevant topic. Authors do not rely on solely the theory or the experiments, but rather bring the two together to build a stronger argument.

**Originality:** Authors propose the idea that flatness of discovered minima correlates with the amount of compression of the inputs. Many studies claim that flatness is correlated with better generalisation, and this paper provides a perspective that may explain this known correlation. Although the idea that compression leads to improved representations is not novel on its own, I have not encountered works that have explicitly linked it to minima flatness.

**Quality and clarity:** The paper is very well-written and easy to follow. The experiments are not extensive, but are well-designed.

**Significance:** I believe the topic of generalisation in NNs to be extremely relevant, thus any insight into the dynamics of training that lead to improved generalisation are significant. The paper brings together a few existing seminal works, and elegantly ties them together. An important contribution is the suggestion to consider parameter space alongside the feature space, which is rarely done in practice.

**Weaknesses:**

The authors consider Hessian eigenvalue magnitudes as the only measure of sharpness. This seems limiting, as recent studies (Yang, L. Hodgkinson, R. Theisen, J. Zou, J. E. Gonzalez, K. Ramchan- dran, and M. W. Mahoney, “Taxonomizing local versus global structure in neural network loss landscapes,”) have shown that Hessian trace may be more closely related to ruggedness of the landscape rather than sharpness of the basin, due to the fact that the Hessian is approximated locally and does not consider the neighborhood of a solution. In fact, the authors state in a few places that sharpness (as measured by the eigenvalues) may not be sufficient to explain the manifold volume reduction.

Experimentation is convincing, although somewhat limited. Only a single CNN architecture is considered. Since fundamental questions are being asked, wouldn’t it make sense to include a standard MLP in the experiments?

**Questions:**

Formatting: equations are referred to as “equation N” and “Eq. (N)” interchangeably. Please use the latter format. Figure references are not always appropriately enclosed in parenthesis.

Placement of figures: Figures 1 and 2 are placed very early in the document, and the reader is expected to “page back” to inspect the figures when they are eventually discussed in the text. While this is a minor inconvenience, I still believe the paper would be more pleasant to read if the figures were placed as close to their first mention in the text as possible.

The paper is written from the standpoint that flatness is an indication of better generalisation. However, this is not a fact, but rather a hypothesis, and should not be treated as an axiom. See, for example, https://arxiv.org/abs/2302.07011, where the relationship between sharpness and generalisation is challenged for modern NN architectures. I would appreciate it if authors could comment on this aspect, and perhaps include a discussion of the open-endedness of sharpness VS generalisation question in the paper.

---

> ### Author Response · Authors · 2023-11-22
> **Thank you**
>
> We thank the reviewer for their critiques and suggestions, and their overall assessment of the value of our central contribution in newly linking flatness and compressed representations.
>
> Below we address the reviewer’s concerns point by point:
>
> Weakness:
>
> We thank the reviewer for pointing out that sharpness may not fully explain volume compression.  To bring this out, in our revision we note in the updated Sec. 3.2 that the bound in Eq. 11 is indeed loose (L162): “We observe that the equality condition in the first line of Eq. (11) rarely holds in practice, since to achieve equality, we need all singular values of the Jacobian matrix to be identical.”. This critique has led us to propose a more precise metric that measures the maximum sensitivity along any direction:  we call this maximum local sensitivity (MLS). We show that a tighter bound can be derived for MLS (Eq. 13), and we empirically test the tightness of the bound in Appendix D.1.  We still note, however, that the volume is positively correlated with sharpness in all of our experiments, so that -- while we acknowledge that significant variability remains to be explained -- sharpness does predict an overall trend that aligns with our main message.
>
> We appreciate the point that more extensive experimentation is needed to make a stronger case.  As the reviewer suggested, we have now added experiments on MLP trained on FashionMNIST dataset (Fig. D.6, E7-8 in the appendix).
>
> Questions:
> Formatting: We thank the reviewer for spotting this. Equations are now consistently referred to as "Eq. (N)," and figure references are now enclosed in parentheses as suggested.
>
> Placement of figures: we agree, and the positions of figures are now adjusted to be close to where they are mentioned
>
> Sharpness vs generalization: We acknowledge the contentious nature of the topic and appreciate this point.  In our revision, we have duly removed or rephrased any overly assertive statement that states sharpness directly leads to generalization. Moreover, we have added substantial additional literature and allied discussion -- including a summary of the mixed and negative results relating sharpness and generalization (second paragraph of Introduction, starting from L37).  Additionally, in Appendix D.1 we newly enumerate conditions that may contribute to such mixed and negative results.  Finally, in Appendix D.2, we empirically test correlations among a wide set of different metrics, further connecting to the broader literature while confirming that our experiments on the VGG10 and MLP networks studied here both do show a positive correlation between sharpness and test loss.

---

### Official Review · Reviewer_Wiq9 · 2023-10-31

**Soundness:** 3 good
**Presentation:** 1 poor
**Contribution:** 2 fair
**Rating:** 3
**Confidence:** 4

**Summary:**

The paper tries to connect loss landscape flatness with generalization, along with compression. It has some theoretical results, and some experiments backing it.

**Strengths:**

The paper studies an interesting avenue of different effects in neural network dynamics, and tries to connect between them in an interesting way. It has some theoretical guarantees, and presents some experimental results.

**Weaknesses:**

** The paper is not well presented. The text is not clear, and the figures are not well presented**
The text has many mistakes, many things are well explained.

Bellow are a few of many comments on the text presentation
1. L73 : The Frobenius norm is defined yet it is not clear for what.
2. L94: "Let W be the input weights to the network, and θ¯ the corresponding set of parameters." - it is not clear what W ("Input Weights") is.
3. L112: This amounts to asking whether the transformation of the differential volume around x¯ is an expansion or a contraction: This is problematic in my opinion, as NN often change the dimensions between layers, and this is not incapsulated in this measure as far as I understand.
4. "The interpolation phase" - is usually used for the phase of zero **error** not zero loss (e.g. Neural Collapse literature).
5. Figure 1 is extremely not clear. The caption is even misleading, as "From left to right" does not include the Train loss.


**Missing Literature:**

[1] Ben-Shaul, I. &amp; Dekel, S.. (2022). Nearest Class-Center Simplification through Intermediate Layers. <i>Proceedings of Topological, Algebraic, and Geometric Learning Workshops 2022</i>, in <i>Proceedings of Machine Learning Research</i> 196:37-47 Available from https://proceedings.mlr.press/v196/ben-shaul22a.html.

[2] Ben-Shaul, I., Shwartz-Ziv, R., Galanti, T., Dekel, S., & LeCun, Y.  Reverse Engineering Self-Supervised Learning. In Proceedings of the Thirty-seventh Conference on Neural Information Processing Systems (NeurIPS 2023).

[3]  Rangamani, A., Lindegaard, M., Galanti, T. &amp; Poggio, T.A.. (2023). Feature learning in deep classifiers through Intermediate Neural Collapse. <i>Proceedings of the 40th International Conference on Machine Learning</i>, in <i>Proceedings of Machine Learning Research</i> 202:28729-28745 Available from https://proceedings.mlr.press/v202/rangamani23a.html.

[4] Gamaleldin F. Elsayed, Dilip Krishnan, Hossein Mobahi, Kevin Regan, and Samy Bengio. Large margin deep networks for classification. In NeurIPS, 2018.

[5] Galanti, T., Galanti, L., & Ben-Shaul, I. (2023). Comparative Generalization Bounds for Deep Neural Networks. Transactions on Machine Learning Research, (ISSN 2835-8856).

**Questions:**

I think adding much more experimental evidence of the effects, explaining them more thoroughly, and well would improve the paper.

---

> ### Author Response · Authors · 2023-11-22
> **Thank you**
>
> We appreciate the reviewer’s critiques and suggestions. Below we address the reviewer’s concerns point by point:
>
> Weakness:
>
> 1. We thank the reviewer for pointing this out -- the Frobenius norm should be defined for the norm in Eq. 3.  We have now moved the explanation to right after Eq. 3.
> 2. Here, “W” are the weights of the linear layer. The setting is the same as Ma & Ying 2021, and as in many deep learning architectures, where the raw features go through a linear layer first. We have updated the text L117 to make this point clearer: “Let W be the input weights (the parameters of the first linear layer) of the network, and (theta bar) the rest of the parameters.”.
> 3. We acknowledge that we are not considering the dimension of layers between the input and the output in our work. However, our work applies when we redefine the function f to be the transformation up until any middle layer that may be of interest. We have updated the text at the beginning of Sec. 3 to reflect this: “Although we only consider the representations of the output of the network, our results apply to representations of any middle layers by defining f to be the transformation from input to the middle layer of interest.”
> 4. We thank the reviewer for pointing out this difference. We have updated the manuscript to note that the solution found by SGD is indeed an “approximate interpolation solution” with zero training error instead of zero training loss due to the gradient noise. Therefore our reference to the interpolation phase in experiments is consistent with those in the neural collapse literature.  In our revision we have also added and proved Lemma A.1, to show that our estimate of the sharpness won’t invalidate our results during the interpolation phase given that the training loss is small enough.
> 5. We apologize for the lack of clarity here -- we have fixed the phrasing in the caption.
>
> Overall, thanks to the reviewer comments the presentation has now been substantially improved.
>
> In addition, we thank the reader for the missing literature, all of which we have cited in the updated version (L24, L29).
>
> Question:
> We thank the reviewer for this suggestion and have added
>
> 1) additional experiments on MLP training on FashionMNIST dataset (Fig. D.4, D.6, E7-8)
> 2) a new experimental appendix D that comprehensively analyzes the tightness of the derived bound in a new figure
> 3) a new analysis, and related figure, for the correlations between different metrics introduced in the paper and accompanying discussion of the connections of our bound to other works.
>
> The contributions of these additions are:
> 1. We identify two representation-space quantities that are bounded by sharpness -- volume compression and the newly added maximum local sensitivity (MLS) -- and give explicit formulas for these bounds (Sec. 3.1, 3.2, Eq. 12-13).
> 2. We conduct empirical experiments with both a VGG and an MLP network and find that volume compression and MLS are indeed strongly correlated with sharpness (Figs D.4-6, and Appendix D.2).
> 3. We find that sharpness, volume compression, and MLS are also correlated, if more weakly, with test error and hence generalization (Figs D.5-6, and also Appendix D.2).
> 4. We carefully inspect the condition when the equality holds for the bounds that we derived, and give a set of conditions that determine when MLS will and will not be tightly correlated with sharpness (Appendix D.1) and, we conjecture, also generalization. We find that when the conditions are (approximately) satisfied, the quantities on both sides of the inequality are highly correlated.
>
> We hope that the reviewer will agree with us that these additions and improvements make a substantially stronger case, and thank them for their role in helping us to make it.

---

### Author Response · Authors · 2023-11-22
**Response to all reviewers, thank you**

We thank the reviewers for their insights and expert knowledge in reviewing our paper.  Each set of reviewer comments has led us to make significant improvements to the revised manuscript, through their suggested enhanced numerical experiments, broader and more modern framing of the paper in the context of more (and now-included) literature, and allied more precise analytical results.  Here we highlight the overall improvements to our paper; we also provide a detailed point-by-point reply to each reviewer.

Following the reviewers' suggestions we improved our paper on all fronts:

Framing:  We have improved the paper's motivation and enhanced and clarified its relationship to prior work.  First, we have deleted or rephrased statements that imply sharpness directly leads to generalization, and have added missing literature -- including a summary of past mixed and negative results relating sharpness and generalization.  Second, we have rearranged central arguments in our paper to clarify what parts form our novel contribution, and have newly included a list of these contributions in the introduction.

Experimentation:  To show how our bound performs in practice and relate it to other metrics in the literature, we have included new numerical results in the new Fig D.4-6, which show how our bound performs in practice.  Moreover, As suggested, we have included new experiments for a general (MLP) architecture in Fig E.7 and E.8 in Appendix E.

Mathematical analysis:  In response to reviewer concerns about the interpolation regime, we have added a new Lemma A.1 which shows that our main results continue to hold in the case of small loss.  We have also removed or rephrased statements that contain “zero loss.” We have also introduced a modified measure of compressed representations, the maximum local sensitivity (Equation 14), and showed that it allows a tighter bound to be derived (Sec. 3.3).

We are confident that these additions and revisions, all following the reviewers’ careful assessments, have strongly improved the original manuscript, and we thank the reviewers again for the suggestions and the opportunity to follow them here.

---

### Meta-Review · Area_Chair_tHae · 2023-12-10

**Metareview:**

This paper studies the relationship between sharpness in the weight space, e.g., the trace of the Hessian of the empirical loss, and the dimensionality of the feature space in a deep network to calculate an upper bound on the volume of the features; the authors argue that the later relates to compression and therefore this paper creates a link between the shape of the energy landscape and the local geometry of the representation space, robustness etc. The reviewers have commented on the clarity/impreciseness of the presentation and insufficient experimental validation. The authors commendably made extensive changes to the manuscript which has improved the paper. But as one of the reviewers said this requires that the paper be reviewed thoroughly again.

Finally, there are actually many papers that have studied the relationship between sharpness and the feature space, e.g., https://arxiv.org/abs/2010.04261, https://arxiv.org/abs/2110.14163 etc.

**Justification For Why Not Higher Score:**

Please see above.

**Justification For Why Not Lower Score:**

N/A

---

### Decision · Program_Chairs · 2024-01-16

Reject